# The unconventional activation of the muscarinic acetylcholine receptor M4R by diverse ligands

Jingjing Wang[1,4], Meng Wu[1,4], Zhangcheng Chen [2,4], Lijie Wu[1], Tian Wang[1,3], Dongmei Cao[2], Huan Wang[1], Shenhui Liu[1,3], Yueming Xu[1], Fei Li[1], Junlin Liu[1], Na Chen[1], Suwen Zhao [1,3], Jianjun Cheng [1✉], Sheng Wang [2✉] & Tian Hua [1,3✉]

Muscarinic acetylcholine receptors (mAChRs) respond to the neurotransmitter acetylcholine and play important roles in human nervous system. Muscarinic receptor 4 (M4R) is a promising drug target for treating neurological and mental disorders, such as Alzheimer's disease and schizophrenia. However, the lack of understanding on M4R's activation by subtype selective agonists hinders its therapeutic applications. Here, we report the structural characterization of M4R selective allosteric agonist, compound-110, as well as agonist iperoxo and positive allosteric modulator LY2119620. Our cryo-electron microscopy structures of compound-110, iperoxo or iperoxo-LY2119620 bound M4R-$G_i$ complex reveal their different interaction modes and activation mechanisms of M4R, and the M4R-ip-LY-$G_i$ structure validates the cooperativity between iperoxo and LY2119620 on M4R. Through the comparative structural and pharmacological analysis, compound-110 mostly occupies the allosteric binding pocket with vertical binding pose. Such a binding and activation mode facilitates its allostersic selectivity and agonist profile. In addition, in our schizophrenia-mimic mouse model study, compound-110 shows antipsychotic activity with low extrapyramidal side effects. Thus, this study provides structural insights to develop next-generation antipsychotic drugs selectively targeting on mAChRs subtypes.

[1] iHuman Institute, ShanghaiTech University, 201210 Shanghai, China. [2] State Key Laboratory of Molecular Biology, Shanghai Institute of Biochemistry and Cell Biology, Center for Excellence in Molecular Cell Science, Chinese Academy of Sciences, 200031 Shanghai, China. [3] School of Life Science and Technology, ShanghaiTech University, 201210 Shanghai, China. [4]These authors contributed equally: Jingjing Wang, Meng Wu, Zhangcheng Chen.
✉email: chengjj@shanghaitech.edu.cn; wangsheng@sibcb.ac.cn; huatian@shanghaitech.edu.cn

Muscarinic acetylcholine receptors (mAChRs) are activated by the important neurotransmitter acetylcholine (ACh) and are involved in a variety of physiological functions[1,2]. Among the five subtypes of muscarinic receptors (M1R–M5R), M1R, M3R, and M5R couple to $G_{q/11}$ protein, while M2R and M4R preferentially signal through $G_{i/o}$ protein[3,4]. M1R and M4R are associated with learning, memory, and cognition[5,6] and are promising targets for the treatment of neurological disorders[7,8]. For example, xanomeline, a M1R/M4R-preferring agonist, is shown to have positive effects on cognitive and psychotic-like symptoms in Alzheimer's disease and schizophrenia[9,10]. However, clinical development is hindered by the dose-limiting adverse effects of xanomeline, which is due to non-selective activation of other peripheral mAChR subtypes[11,12].

Structure and sequence analysis indicates that the major hurdle in developing mAChR subtype-selective agonists is the high homology of their orthosteric binding pockets. Fortunately, the paired orthosteric and allosteric sites have been found to exist in muscarinic receptors. The less conserved allosteric binding pockets are supposed to provide an opportunity for developing allosteric drugs that are more subtype-selective[13,14]. Several selective positive allosteric modulators (PAMs) for M4R show central nervous activity and preclinical efficacy[15–17]. However, some problems still remain unsolved in creating mAChR subtype selective agonists and PAMs. For example, LY2119620 non-selectively modulates both M2 and M4 receptors cooperatively with the potent muscarinic-agonist iperoxo.

In this study, a functionally characterized M4R-preferring allosteric agonist, compound-110, as well as orthosteric agonist and PAM, are investigated. Our cryo-EM structures of compound-110, iperoxo and iperoxo-LY2119620 activated M4R in complex with heterotrimeric $G_i$ protein unveil the binding modes of diverse agonists and PAM as well as activation mechanism of M4R. Of note, we further investigated the therapeutic potential of compound-110 on schizophrenia-mimic mouse model, and the results showed that compound-110 has high brain penetrability and antipsychotic activity. This study may promote the development of neurological disorder drugs on muscarinic receptors with safer pharmacological profile.

## Results

**Cryo-EM structures of M4R–$G_i$ in complex with different agonists and PAM.** Compound-110 (provided by Merck) is a M4R agonist which possesses a bipiperidine scaffold attached to a benzo[d]imidazol-2-one core, and is structurally similar to M1R agonists TBPB and compound-12a[18–20] (Supplementary Fig. 1a). However, as a TBPB's ethyl carbamate analog, compound-12a displays a potent agonist with moderate selectivity on M4R[20]. From a structure–activity relationship (SAR) point of view, the carbamate tail in compound-12a plays an important role in activating M4R. Compared to compound-12a, compound-110 bears a similar methyl carbamate tail, with an additional methyl substituent attached to the benzo[d]imidazol-2-one core. Thus, it appears that the methyl substituent plays a critical role in enhancing the M4R selectivity of compound-110 (Supplementary Fig. 1a). In addition, compound-110 was characterized as an agonist with slightly higher potency at M4R than that at M2R and M1R in cAMP and $Ca^{2+}$ mobilization assays (Supplementary Fig. 1b–d). Since the agonist potency is affected by the system in which it is measured, further investigation of the compound-110's selectivity in different assays[21] is needed.

To obtain the agonist-bound or agonist and PAM-bound M4R in complex with $G_i$, we co-expressed the receptor and $G_i$ proteins in insect cells. Eventually, we were able to improve the expression level of M4R by inserting BRIL protein at the N terminus, and the

stable complex was successfully assembled by removing residues K240-N372 of ICL3. The M4R–compound-110–$G_i$–scFv16 (M4R–c110–$G_i$) complex structure was obtained at a nominal global map resolution of 3.6 Å through single-particle cryo-EM analysis (Fig. 1a and Supplementary Fig. 2). Furthermore, the cryo-EM structures of M4R–iperoxo–$G_i$–scFv16 (M4R–ip–$G_i$) and M4R–iperoxo–LY2119620–$G_i$–scFv16 (M4R–ip–LY–$G_i$) were also solved at resolutions of 3.4 Å (Fig. 1b, Supplementary Fig. 3, Supplementary Table 1).

The overall structures of the three complexes are similar, with the root mean square deviations (RMSDs) of the Cα atoms of receptors are around 0.9 Å. Compared with the M4R–ip–$G_i$ structure, the extracellular loop 3 (ECL3) as well as the extracellular part of TM7 in M4R–ip–LY–$G_i$ and M4R–c110–$G_i$ structures, are closer to ECL2 and thus cause more contraction of the extracellular vestibule, while the intracellular parts of TM6 show more outward movement (Supplementary Fig. 4a). The diverse scales of TM6's outward movement may lead to the orientation differences of $G\alpha_i$ subunits in those complex structures, where M4R–ip–$G_i$, M4R–ip–LY–$G_i$, and M4R–c110–$G_i$ structures show the outward movement of 10.5, 11.4, and 12.3 Å, respectively, compared with the inactive-state M4R structure (Supplementary Fig. 5a, 10b).

In general, the binding interfaces of the three M4R and $G_i$ complexes are similar to that of M2R–$G_{oA}$ complex (Supplementary Fig. 5b, d), where additional salt bridges are observed in the M4R–$G\alpha_i$ subunit interface (Supplementary Fig. 5c). In addition, $Val398^{6.33}$ (Ballesteros–Weinstein numbering system)[22] forms hydrophobic interactions with the highly conserved $Leu353^{G.H5.25}$ and $Leu348^{G.H5.20}$ (CGN numbering system)[23] of $G_i$ protein in our M4–$G_i$ complex structures, which further support the observation that residues $V^{6.33}T^{6.34}xxI^{6.37}L/F^{6.38}$ in the intracellular part of TM6 are important for M2R and M4R coupling with $G_{i/o}$ (Supplementary Fig. 5e).

**Iperoxo and LY2119620 binding pockets in M4R.** Though iperoxo is a highly efficacious agonist for M4R and other mAChR subtypes[24,25], the EM density is not clear enough for an unambiguous placement of iperoxo in the binding pocket of the M4R–ip–$G_i$ structure. However, the density for iperoxo is much improved in the M4R–ip–LY–$G_i$ structure, indicating that LY2119620 stabilizes the binding of iperoxo with M4R (Supplementary Fig. 3f). Concordantly, a similar iperoxo binding pose is reported in both M2R–ip–LY–$G_{oA}$ and M1R–ip–$G_{11}$ structures owing to the highly conserved residues in the orthosteric binding pockets of mAChRs. In the M4R–ip–LY–$G_i$, M2R–ip–LY–$G_{oA}$ and M1R–ip–$G_{11}$ structures, the conserved residues that form interactions with iperoxo are $Asp^{3.32}$, $Tyr^{3.33}$, $Tyr^{6.51}$, $Asn^{6.52}$, $Tyr^{7.39}$, and $Tyr^{7.43}$ (Supplementary Fig. 6a–c). Of note, the surface-accessible volume of the orthosteric binding pocket in M4R (99 Å³) is larger than that of M2R (82 Å³) (as calculated by POVME2.0)[26] (extended data Fig. 6d–g). This might contribute to the higher affinity of iperoxo with M2R than M4R[27,28]. Compared to the crystal structures of inactive M4R[29,30], the interaction network among $Asp112^{3.32}$, $Ser85^{2.57}$, $Trp108^{3.28}$, $Tyr443^{7.43}$, and the "tyrosine lid" formed by three conserved tyrosine residues at positions 3.33, 6.51, and 7.39 in mAChRs, induces a smaller agonist-binding pocket than that of a larger antagonist binding pocket with volume of 185 Å³ (Fig. 2a and Supplementary Fig. 6d).

In the PAM binding site, $Trp435^{7.35}$ and $Phe186^{ECL2}$ stack into the sandwich-like π–π interactions with LY2119620 (Fig. 2b), which is consistent with the functional and computational studies on its analog LY2033298[28,31,32]. Although the binding pose of LY2119620 in M4R is similar to that in M2R, some differences are

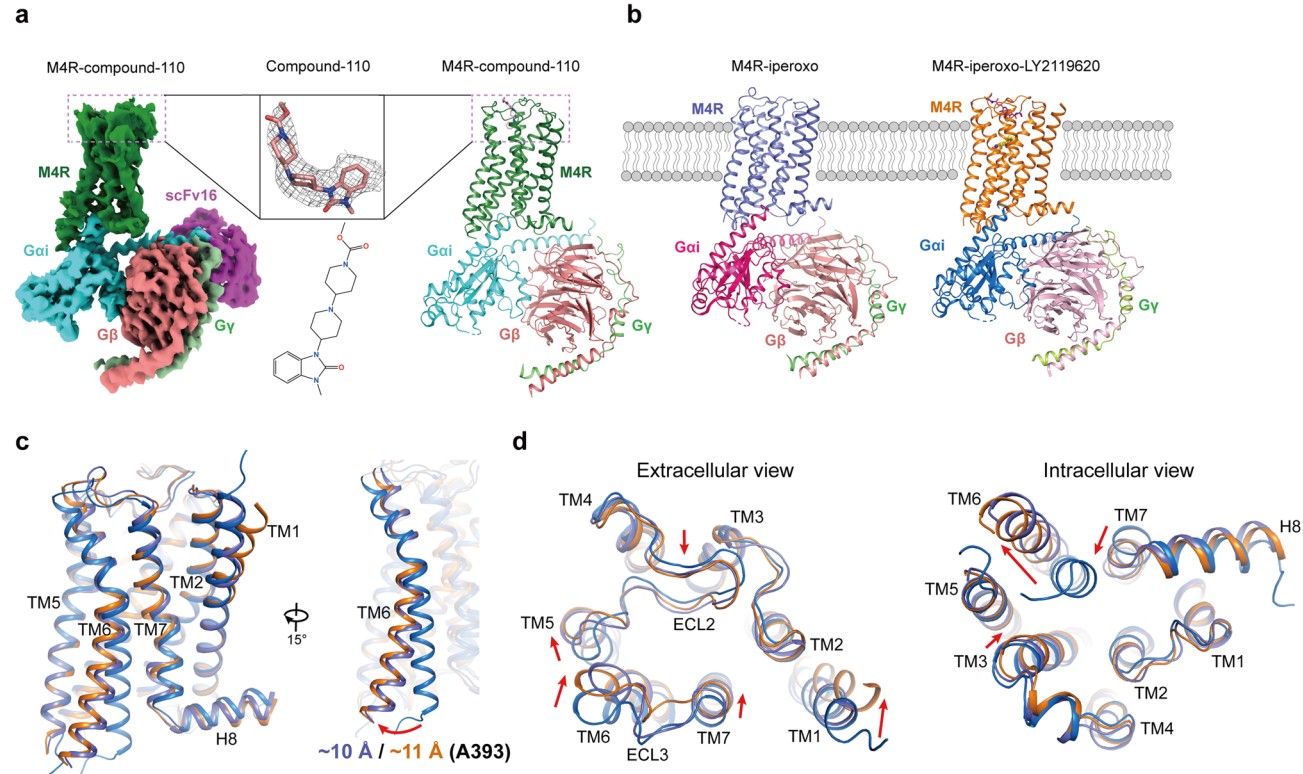

**Fig. 1 Cryo-EM structures of the M4R complexes. a** Cryo-EM map of M4R–Gᵢ–scFv16 in complex with compound-110 (left) and cartoon representation of the M4R–c110–Gᵢ complex structure (right). The cryo-EM density of compound-110 (salmon) and the two-dimensional representation of the compound-110 chemical structure is shown. **b** The cryo-EM structures of M4R–ip–Gᵢ complex (left) and M4R–ip–LY–Gᵢ (right) complex are shown in cartoon representation. Iperoxo and LY2119620 are shown as yellow ball-sticks and magenta sticks, respectively. **c** The outward movement of TM6 in M4R–ip–Gᵢ and M4R–ip–LY–Gᵢ structures compared with that in inactive M4R structure (PDB code 5DSG), with residue A393 as reference. **d** Extracellular and intracellular views of inactive and active M4R structures. Conformational changes from inactive to active state are indicated with red arrows.

also observed. For example, the piperazine ring of LY2119620 in M4R points horizontally toward the ECL2 near TM5 and ECL3, while it orientates vertically forming a salt bridge interaction with Glu172$^{ECL2}$ in M2R. Interestingly, the corresponding residue in ECL2 is Pro181 in M4R and it does not form interactions with LY2119620 (Fig. 2c). The different shape and distribution of charged residues in the allosteric sites of M2R and M4R may lead to the subtle difference of the binding pose of LY2119620 between the two receptors (Fig. 2d).

**Activation mechanism of M4R by iperoxo and LY2119620.** Comparing the inactive and active structures of M4R, the most significant conformational differences occur in TM5, TM6, and ECL3. During activation, the extracellular part of TM6 bends into the core of the receptor resulting in the inward shift of ECL3, and TM5 rotates towards TM4, while the intracellular part of TM6 swings out (Fig. 1b, c). In the orthosteric binding pocket of M4R–ip–LY–Gᵢ structure, Asp112$^{3.32}$ forms the charge interaction with the quaternary ammonium of iperoxo, while Tyr113$^{3.33}$, Tyr416$^{6.51}$, and Tyr439$^{7.39}$ form cation–π interactions with iperoxo (Supplementary Fig. 6a). In addition, Asn417$^{6.52}$ forms a hydrogen bond with iperoxo, which also appears in the M2R and M1R structures. Those interactions and conformational changes induce the rotation of Trp413$^{6.48}$ and the synergetic rearrangement of PI(V)F transmission switch (Val120$^{3.40}$, Phe409$^{6.44}$, and Pro207$^{5.50}$), which is followed by the outward movement of the cytoplasmic part of TM6 (Supplementary Fig. 7a). The activation process of M4R also exhibits other characteristics of class A GPCR activation, where Arg130$^{3.50}$ in the DRY motif adopts an extended rotation to potentially form a hydrogen bond with

Tyr453$^{7.53}$ of the NPxxY motif, thereby inducing the partial "unwinding" of TM7 and enhancing TM3–TM7 packing (Supplementary Fig. 7b, c).

In the M4R–ip–LY–Gᵢ structure, the thienopyridine ring of LY2119620 reorients the sidechains of Trp435$^{7.35}$ and Phe186$^{ECL2}$ and forms π–π interactions to tether the extracellular portion of M4R into a more compact conformation (Figs. 2b, 1d). The scale of the activation-induced closure of the allosteric vestibule and that of the outward movement of the cytoplasmic end of TM6 vary in descending and ascending orders, respectively, in the M4R–ip–Gᵢ and M4R–ip–LY–Gᵢ structures. For example, TM6 swings out about 10 and 11 Å (Cα atom of Ala393 as the reference) in those two structures, separately (Fig. 1b, c).

Interestingly, the conformational change of Trp435$^{7.35}$ also influences the interaction network of the "tyrosine lid", especially the rotation of Tyr439$^{7.39}$. The tyrosine lid formed by Tyr113$^{3.33}$, Tyr416$^{6.51}$, and Tyr439$^{7.39}$ in the active M4R structures looks like an unscrewed half-opened bottle cap, instead of the closed, flat tyrosine lid in the active M2R structure (Fig. 2c, d). This may be due to the lower cooperativity between iperoxo and LY2119620 in M4R. Previous data showed that the iperoxo and LY2119620 pairing has a cooperativity with α factor of 3.9 in M4R and 14.5 in M2R[28]. This indicates that allosteric modulators are capable of stabilizing receptors into different subtype-specific active states even with the same orthosteric ligand. For a better understanding of the allosteric modulation or cooperative between iperoxo and LY2119620, we performed MD analysis of M4R with different ligands. The results show that iperoxo has lower flexibility in the M4R–ip–LY structure compared with that of M4R-ip structure (Supplementary Fig. 8a, b).

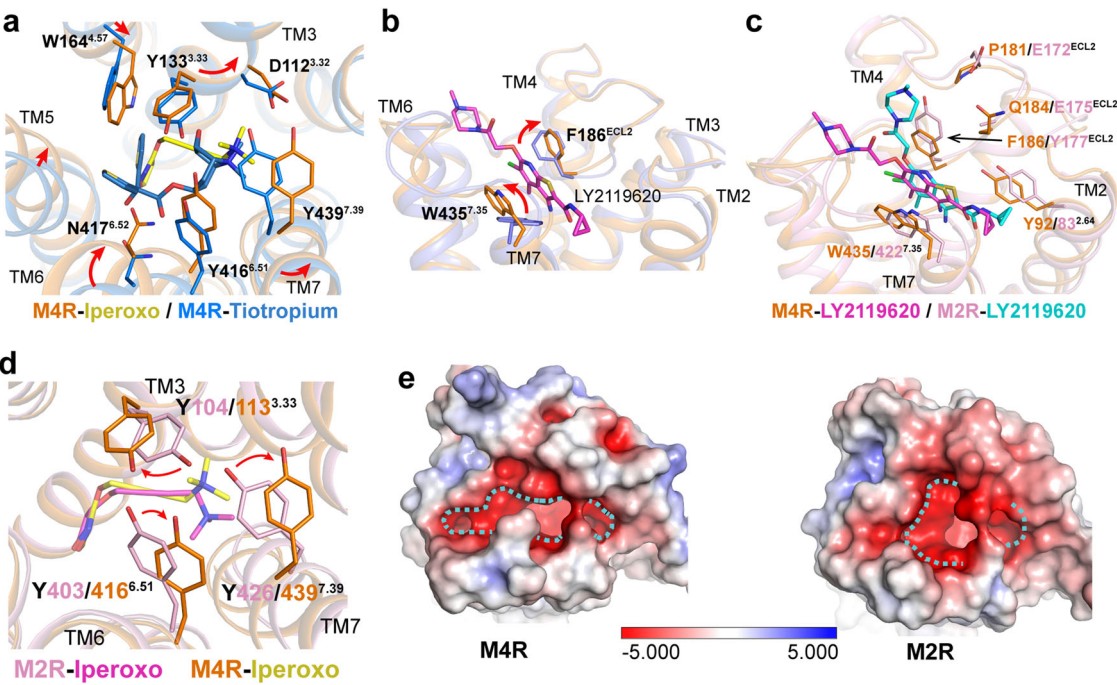

**Fig. 2 Comparison of iperoxo and LY2119620 binding pockets in M4R and M2R structures. a** Conformational changes of key residues within the orthosteric binding pocket in inactive M4R (PDB code 5DSG, blue) and active M4R (orange) structures. **b** Rotation changes of residues F186$^{ECL2}$ and W435$^{7.35}$ in the M4R–ip–LY–G$_i$ structure to form π–π interactions with LY2119620 compared to that of in M4R–ip–G$_i$ structure. **c** Comparison of LY2119620 binding pose, and **d** tyrosine lid between active M4R–ip–LY–G$_i$ and M2R–ip–LY–G$_{oA}$ (PDB code 6oik, pink) structures. **e** Electrostatic and shape properties of the LY2119620 binding site in active M2R and M4R structures. Negatively charged residues are colored as red. Dotted line delineates the shape of the allosteric binding pockets.

**The binding mode of compound-110 in M4R.** In the M4R–c110–G$_i$ structure, compound-110 mostly occupies allosteric binding pocket with vertical binding pose, instead of the more horizontal layout of PAM LY2119620. Also, the benzoimidazolone of compound-110 points deeper into the "tyrosine lid" and its middle piperidine overlaps with the bicyclic core of LY2119620 in the M4R–ip–LY–G$_i$ structure (Fig. 3a). However, compound-110 does not show any PAM activity in the measurements of Gα$_{i/o}$-βγ dissociation and β-arrestin recruitment in our optimized bioluminescence resonance energy transfer (BRET) assay (Fig. 3d, Supplementary Fig. 9, Supplementary Table 2). Thus, we redefine compound-110 as an allosteric agonist of M4R.

The main interacting residues within 4 Å around compound-110 are Tyr92$^{2.64}$, Phe186$^{ECL2}$, Tyr416$^{6.51}$, Val420$^{6.55}$, Asp432$^{ECL3}$, Trp435$^{7.35}$, and Ser436$^{7.36}$. Among them, the Trp435$^{7.35}$ and cation nitrogen in the middle piperidine ring of compound-110 form a cation–π interaction, which likely stabilizes the binding of compound-110. Compound-110 binds around 1.6 Å deeper vertically than LY2119620 and suppresses the side chain of Tyr439$^{7.39}$, causing it to point downward into the orthosteric binding pocket (Fig. 3c, Supplementary Fig. 10a). Consistently, the W435$^{7.35}$A and Y439$^{7.39}$A mutations dramatically attenuate compound-110's agonist potency and efficacy (Fig. 3b and Supplementary Fig. 9). Moreover, the distance between Phe186$^{ECL2}$ and compound-110 is quite far and does not form the sandwich-like π–π interaction that is observed in PAM LY2119620's case. Accordingly, the F186$^{ECL2}$A mutation slightly increases the potency of compound-110 in both cAMP and BRET assays (Fig. 3b and Supplementary Fig. 9).

**The activation features of M4R by compound-110.** It is intriguing to know how an allosterically bound compound-110 could

fully activate M4R. Firstly, the extracellular vestibule in M4R–c110–G$_i$ structure is similar to that of the M4R–ip–LY–G$_i$ structure (Supplementary Fig. 10b, c). However, in contrast to the semi-horizontal-bound LY2119620 and sandwich like π–π interaction with both Trp435$^{7.35}$ and Phe186$^{ECL2}$, compound-110 takes a more vertically deeper binding pose that forms cation–π interaction with Trp435$^{7.35}$ and a salt bridge with Asp432$^{ECL3}$. This configuration induces the inward movement of ECL3 and the extracellular vestibule around TM5, TM6, and TM7 (Supplementary Fig. 10d). Of note, Asp432$^{ECL3}$ forms a hydrogen bond with Tyr92$^{2.64}$, which further stabilizes the active conformation of the extracellular region of M4R (Fig. 4a, b and Supplementary Fig. 10d). Synergistically, those conformational changes and deeper binding of compound-110 cause the rearrangement of the conserved "tyrosine lid", where the sidechains of Tyr439$^{7.39}$ and Tyr113$^{3.33}$ point downward to the orthosteric binding pocket and partially fill the space otherwise occupied by iperoxo in the M4R–ip–LY–G$_i$ structure (Figs. 3a, 4a and Supplementary Fig. 10e). Consequently, three residues, Asp112$^{3.32}$, Tyr443$^{7.43}$, and Ser85$^{2.57}$, are confined to form a special 'triangle frame' that stabilizes the conformation of TM2, TM3, and TM7 (Fig. 4c). In addition, compound-110 directly interacts with Tyr416$^{6.51}$, through which the activation signal is propagated to the transmission microswitch, Trp413$^{6.48}$ (Fig. 4d). In our MD simulation of M4R–c110 structure, Tyr416$^{6.51}$ is observed to maintain a relatively stable distance with Trp413$^{6.48}$, which facilitates a stable interaction (Supplementary Fig. 11). Furthermore, the conformational changes of the key residues mentioned above, coupled with the rotational movement of extracellular parts of TM5 and TM6, induce the flip of the sidechain of Trp413$^{6.48}$, followed by a 12.7 Å outward swing of cytoplastic part of TM6 (Cα atom of Ala393 as the reference) (Supplementary Fig. 10b). Eventually, the activation process of M4R by compound-110 also exhibits the classical

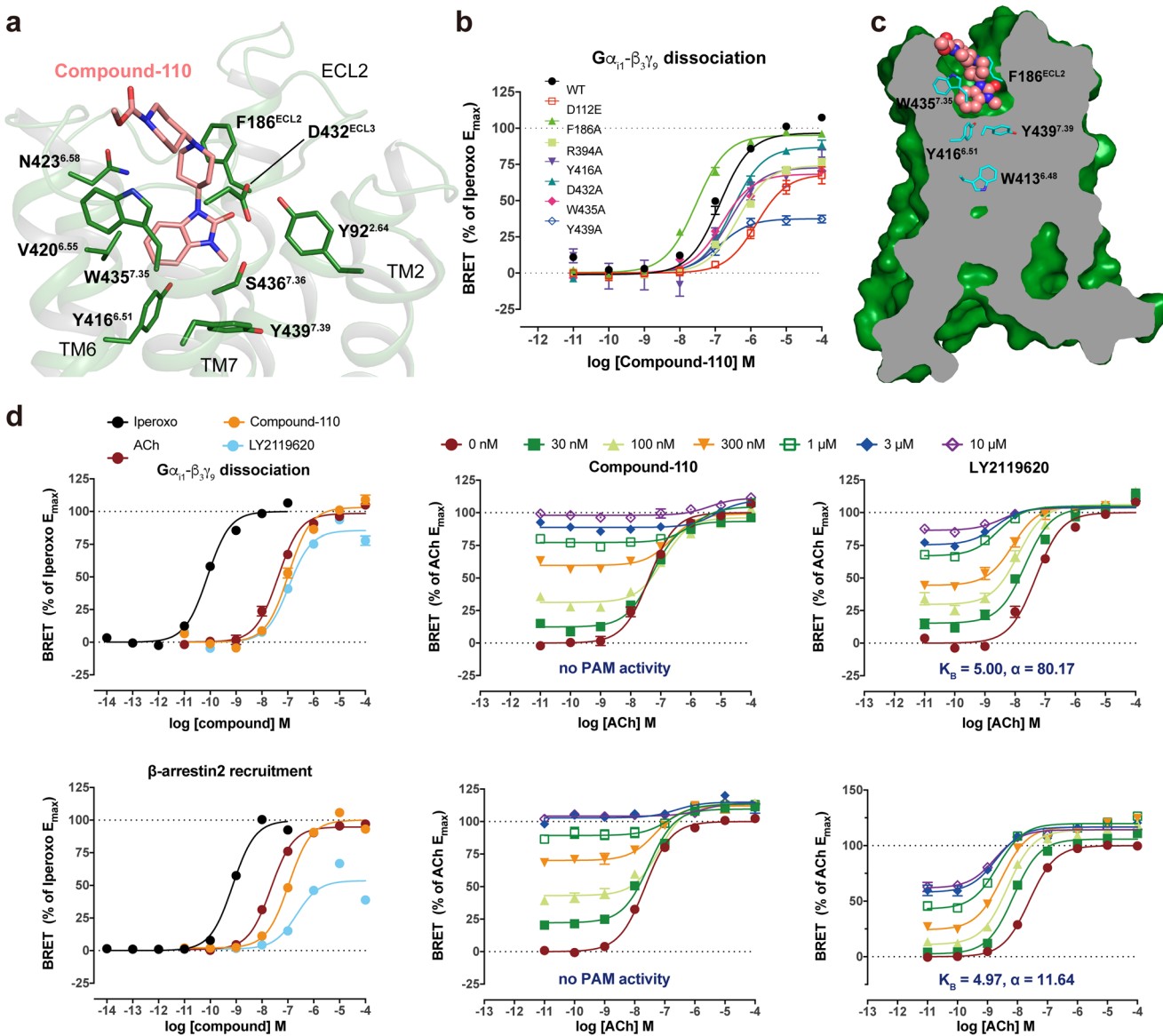

**Fig. 3 Allosteric agonist compound-110 binding mode with M4R. a** Detailed interactions between compound-110 (pink) and M4R (green) from the extracellular view. Residues involved in the binding pocket of M4R are mainly hydrophobic (green sticks) and are derived from TM2, TM6, TM7, and ECL3. **b** The BRET results of WT-M4R and mutants in coupling with $G_{i1}$. Values are shown as the mean ± SEM from $n = 4$ independent biological replicates, each biological replicate has two technical replicates. **c** The side view of the compound-110 (pink sphere) binding pose in M4R (green) with the key residues (cyan) related to activation. **d** The characterization of compound-110 in BRET assay. Source data are provided as a Source Data file.

activation characteristics, including the conformational change of DRY and NPxxY motifs (Fig. 4a, e).

Accordingly, the D432[ECL3]A mutation decreased the potency of compound-110 while not affecting the potency of iperoxo, ACh, or LY2119620 in the BRET assay (Fig. 3b, Supplementary Fig. 9 and Supplementary Table 2). In contrast, the Y439[7.39]A mutation led to a 60% efficacy reduction for compound-110, yet about a 1000-fold decrease in potency for iperoxo or ACh (Supplementary Table 2). The mutation studies further illustrate that compound-110 initiates the activation of M4R via the interaction network which is different from the canonical paired agonist-PAM activation.

## The investigation of antipsychotic potency of compound-110 in schizophrenia-mimic mouse model.
Previous studies suggested that agonists with selective affinity for M4R provide potentially therapeutics to treat schizophrenia[7,10]. Our pharmacokinetics

assay showed that compound-110 exhibited favorable brain-penetrate performance (Fig. 5a), then, we implemented locomotion test to examine potential antipsychotic activity of compound-110. The MK-801(Dizocilpine)-induced hyperlocomotion is a widely used positive symptomatic indicator in schizophrenia-mimic mouse model[33]. Fortunately, as shown in Fig. 5b, compound-110 dose-dependently decreased MK-801-induced hyperlocomotion, with an antipsychotic-like potency (ED50 = 0.06 mg/kg) (Fig. 5c), which is significantly higher than that of VU0467154[34], a potent M4R-positive allosteric modulator. Of note, typical antipsychotic haloperidol was inclined to induce catalepsy, whereas high dose of compound-110 (10 mg/kg) did not induce catalepsy, indicative of low liability of extrapyramidal side effects (Fig. 5d). During the animal experiments, we observed remarkable hypothermia symptoms in mice group injected with high dose (1 mg/kg) of compound-110. We speculate that this side effect may be caused by off-target activation of M2R as previously reported[12,35].

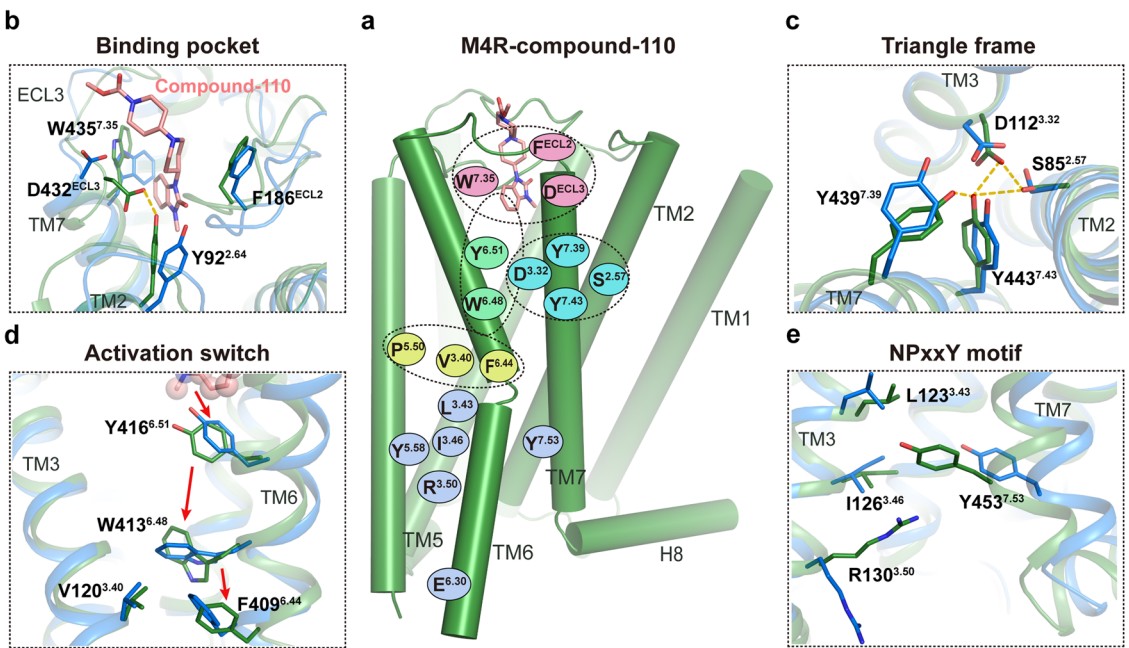

**Fig. 4 The unconventional activation mechanism of compound-110. a** Schematic summarizing the key translational and rotational movements contributing to M4R activation induced by compound-110. **b** Extracellular conformational changes from inactive (blue) to compound-110-bound active (green) M4R. **c** A specific 'triangle frame' formed by the side chains of residues Asp112[3.32], Ser85[2.57], and Tyr443[7.43] in the M4R–c110–G$_i$ structure. **d** Conformational changes of residues Tyr416[6.51] and Trp413[6.48] during M4R activation by compound-110. **e** Conformational changes of the NPxxY motif between inactive and compound-110-bound active M4R structures.

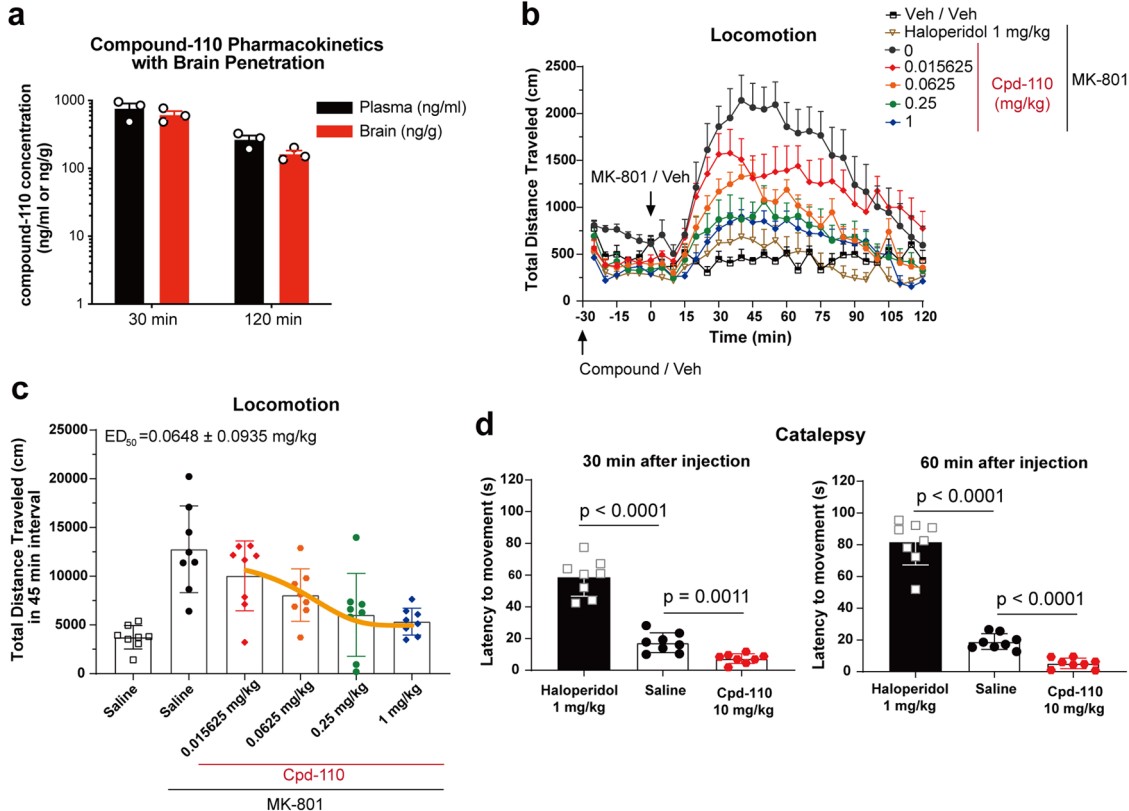

**Fig. 5 Compound-110 reverses MK-801-induced hyperlocomotion without inducing catalepsy. a** Brain penetration of compound-110 in mice ($n = 3$) after 1 mg/kg compound-110 intraperitoneally (i.p.) administration. **b** i.p. administration of compound-110 dose-dependently reverses MK-801-induced hyperlocomotion. **c** The median effective dose (ED$_{50}$) of compound-110 is calculated from total traveled distance in 0–45 min interval in **a** after MK-801 administration using 'one-phase-exponential decay' method. **d** Compound-110 does not elicit recognizable catalepsy; haloperidol (1 mg/kg) is used as the positive control to elicit catalepsy as assessed by latency to movement, $p$-value is calculated using unpaired $t$-test. Data are shown as mean ± SEM of 8 animals per group in **b**–**d**. Source data are provided as a Source Data file.

## Discussion

Here, we report three structures of orthosteric agonist iperoxo bound M4R-$G_i$ in complex with/without PAM, as well as an allosteric agonist compound-110 bound M4R–$G_i$. Through comprehensive structure analysis, this study reveals the active conformations of M4R stabilized by diverse agonists and PAM. The M4R–ip–LY–$G_i$ structure validates the cooperativity between iperoxo and LY2119620 on M4R, in which more tightened extracellular vestibule and clearer electron density of iperoxo in M4R–ip–LY–$G_i$ structure than that in M4R–ip–$G_i$ structure are observed. Additionally, the visible differences of LY2119620 binding in M4R and M2R are analyzed, which may guide the structure-based design of selective allosteric drugs for mAChR subtypes.

Of note, this study also uncovers the unique binding pose of allosteric agonist compound-110 in M4R. The vertical binding pose and the downward intruding benzoimidazolone group of compound-110 facilitate both selectivity and agonism on M4R. Though the selectivity pharmacology profile of compound-110 among muscarinic receptor subtypes is not ideal, it is the only reported ligand that mainly occupies the allosteric binding pocket, yet exhibits unconventional activation features in M4R. We found that compound-110 behaves like a bitopic ligand, where it mostly overlaps with LY2119620 and stabilizes the extracellular vestibule of receptor, while its benzoimidazolone group disrupts the "tyrosine lid" and initiates the activation process. More importantly, compound-110 shows efficacious antipsychotic activity without generating catalepsy, which holds high translational potential for the treatment of schizophrenia. In general, the allosteric agonist with unique binding mode and activation profile should provide hints for the development of safer neurological disorder drugs for the muscarinic receptors.

## Methods

**Data analysis and figure preparation**. Figures were created using the PyMOL 2.3.4 Molecular Graphics System (Schrödinger, LLC), and the UCSF Chimera X 0.9 package. Data were performed using Prism 8.1 (GraphPad).

**Synthesis and characterization of compound-110**

*General*. Commercial reagents and solvents were used as obtained without further purification. $^1H$ NMR and $^{13}C$ NMR spectra were recorded on Bruker AVANCE-III spectrometer at 800 and 201 MHz, respectively. High-resolution mass spectra (HRMS) were measured using an LCMS-IT-TOF (Shimadzu) mass spectrometer in ESI mode. Compound purity was determined by analytical HPLC (Shim-pack GIST C18 column (250 × 4.6 mm, particle size 5 μm) with 0.05% TFA in $H_2O$/0.05% TFA in MeOH gradient eluting system.

Commercially available reagents 1-methyl-3-(piperidin-4-yl)-1,3-dihydro-2H-benzo[d]imidazol-2-one (CAS# 53786-10-0, 1.0 eq) and methyl 4-oxopiperidine-1-carboxylate (CAS# 29976-54-3, 1.0 eq) were dissolved in MeOH, and NaB(CN)$H_3$ (2.0 eq) was added. The mixture was stirred at room temperature overnight. Volatiles were removed under reduced pressure and the residue was purified on flash chromatography (10–20% methanol in dichloromethane) to give **compound-110** as a light-yellow solid. HPLC purity: 98.2% ($\lambda$ = 280 nm).

$^1H$ NMR (800 MHz, CDCl$_3$) δ 7.30 (d, $J$ = 7.2 Hz, 1H), 7.09–7.04 (m, 2H), 6.97 (dd, $J$ = 8.0, 1.6 Hz, 1H), 4.39–4.35 (m, 1H), 4.25–4.15 (m, 2H), 3.70 (s, 3H), 3.41 (s, 3H), 3.08–3.05 (m, 2H), 2.80–2.76 (m, 2H), 2.58–2.54 (m, 1H), 2.44–2.40 (m, 4H), 1.85–1.80 (m, 4H), 1.51–1.47 (m, 2H). (Supplementary Fig 12).

$^{13}C$ NMR (201 MHz, CDCl$_3$) δ 155.92, 154.05, 130.23, 128.20, 121.03, 120.90, 109.64, 107.56, 62.03, 52.66, 51.29, 48.98, 43.65, 29.67, 28.07, 27.19. (Supplementary Fig 13).

HRMS (ESI) $m/z$ calculated for $C_{20}H_{29}N_4O_3^+$ ([M + H]$^+$): 373.2234; found: 373.2237 (Supplementary Fig 14).

**Construct design and expression of M4R and $G_i$ heterotrimer**. Human M4R containing N-terminal thermostabilized apocytochrome b562RIL (BRIL) fusion protein and a deletion of residues 240–372 at ICL3 was cloned into a modified pFastBac1 vector, with the N-terminal haemagglutinin signal peptide (HA) followed by a 10 × His tag, a Flag tag and the HRV-3C cleavage site. Then M4R and human $G_i$ heterotrimer (G$\alpha_{i1}$ and G$\beta_1\gamma_2$) were co-expressed in *Spodoptera frugiperda* (Sf9) super3 insect cells at a cell density of $2 \times 10^6$ cells/ml using the Bac-to-Bac Baculovirus Expression System (Invitrogen). The insect cells were co-infected with three separated viruses for M4R, G$\alpha_{i1}$, and G$\beta_1\gamma_2$ at a stoichiometry ratio 2:1:1

with P1 virus at a multiplicity of infection (MOI) of 5. Cells were harvested after 48 h post infection at 27 °C, and collected by centrifugation. The cell pellets were stored at −80 °C for future use.

**M4R–$G_i$–scFv16 complex purification**. The workflows for M4R–$G_i$–scFv16 complex bound with different ligands were similar, except for adding specific ligand with its own concentration in the lysis, solubilizing, and purification buffers. The final concentrations of iperoxo, LY2119620 and compound-110 were 20, 337, 15 μM separately, for M4R–ip–$G_i$ complex, M4R–ip–LY–$G_i$ complex, and M4R–c110–$G_i$ complex formation. The detail protocol of M4-c110-$G_i$ complex was described as below. The *sf9_super3* insect cell pellets corresponding to 2 L M4R-$G_i$ co-expression culture were thawed and lysed in the hypotonic buffer of 10 mM HEPES pH 7.5, 10 mM MgCl$_2$, 20 mM KCl with EDTA-free cOmplete protease inhibitor cocktail tablets (Roche). The M4R-$G_i$ complex was formed in membranes by addition of 15 μM compound-110 and 2 units of apyrase (NEB) in the presence of 300 μg scFv16. The lysate was incubated at 4 °C for 6 h and discarded the supernatant by centrifugation at 40,000 rpm for 40 min. The M4-c110-$G_i$ complex in the membrane was solubilized using the buffer containing 30 mM HEPES pH 7.5, 500 mM NaCl, 0.75% (w/v) lauryl maltose neopentyl glycol (LMNG, Anatrace), 0.075% (w/v) cholesterol hemisuccinate (CHS, Sigma-Aldrich), 15 μM compound-110 and 2 units of apyrase (NEB) at 4 °C for 2.5 h. The supernatant was isolated by ultracentrifugation, and then incubated with TALON IMAC resin (Clontech) and 20 mM imidazole overnight at 4 °C. The column was washed with 20 CV (column volumes) of washing buffer I containing 20 mM HEPES pH 7.5, 500 mM NaCl, 10% (v/v) glycerol, 0.03% (w/v) LMNG, 0.003% (w/v) CHS, 30 mM imidazole and 15 μM compound-110, and 20 CV of washing buffer II containing 20 mM HEPES pH 7.5, 200 mM NaCl, 10% (v/v) glycerol, 0.01% (w/v) LMNG, 0.001% (w/v) CHS, 50 mM imidazole and 15 μM compound-110. The protein was eluted by 3 CV of elution buffer containing 20 mM HEPES pH 7.5, 100 mM NaCl, 2% (v/v) glycerol, 0.01% (w/v) LMNG, 0.001% (w/v) CHS, 250 mM imidazole, 15 μM compound-110. The purified M4R–c110–$G_i$ complex was concentrated and added 100 μg scFv16 protein, and then incubated at 4 °C for 2 h. Finally, the sample was further purified by size-exclusion chromatography, using Superdex200 10/300 GL column (GE Healthcare) equilibrated with the buffer containing 20 mM HEPES pH 7.5, 100 mM NaCl, 0.00075% (w/v) LMNG, 0.00025% GDN, 0.00001% (w/v) CHS. The complex peak fractions were collected and concentrated individually to 1.5–2.2 mg/ml for cryo-EM sample preparation. 15 μM compound-110 (400 μM iperoxo and 337 μM LY2119620 for M4R–ip–LY–$G_i$, and 400 μM iperoxo for M4R–ip–$G_i$) was added during concentration.

The construction, expression and purification of scFv16 were performed as previously described[36].

**Cryo-EM sample preparation and data collection**. Three microliters of purified M4R–Gi–scFv16 complex were applied to a glow-discharged holey carbon grid (CryoMatrix Amorphous alloy film R1.2/1.3, 300 mesh), and subsequently vitrified using a Vitrobot Mark IV (Thermo Fisher Scientific). The sample was blotted for 3.0 s with blot force 2, within the Vitrobot chamber of 100% humidity at 4 °C. Cryo-EM images were collected on a Titan Krios microscope operated at 300 kV equipped with a Gatan Quantum energy filter, with a slit width of 20 eV, a Gatan K2 summit direct electron camera (Gatan). Movies were taken in EFTEM nanoprobe mode, with 50 μm C2 aperture, at a calibrated magnification of 130,000 corresponding to a magnified pixel size of 1.04 Å. Each movie comprises 40 frames with a total dose of 60 electrons per Å$^2$, exposure time was 8.1 s with the dose rate of 8e$^-$/Å$^2$/s. Data acquisition was done using SerialEM software[37] with a defocus range of −1.2 to −2.0 μm.

**Cryo-EM data processing**. For the M4R–c110–$G_i$–scFv16 complex, 6665 movies were collected and analyzed with cryoSPARC v.2.11[38]. Beam-induced motion correction was performed using patch motion correction. Contrast transfer function (CTF) parameters for each dose-weighted micrograph were estimated by patch CTF estimation in cryoSPARC. A total of 5,125,104 particles were autopicked, and used to do two cycles of 2D classification, then a total of 421,790 particle projections were selected to construct four classes of initial models and used as initial reference models for the subsequent several rounds of 3D classification in cryoSPARC. The final dataset of 78,471 particle projections from the best class was further applied for final homogenous refinement in cryoSPARC, a final density map was obtained with nominal resolution of 3.6 Å (determined by gold standard Fourier shell correlation (FSC) using the 0.143 criterion). Estimation of local resolution was performed with the Bsoft package[39] using the two unfiltered half-maps.

For M4R–ip–$G_i$–scFv16 and M4R–ip–LY–$G_i$–scFv16 complexes, the particles were 4,814,856 and 2,222,068, separately, in 2D classification and 1,477,308, 389,597 particles for the final 3D reconstruction, respectively.

**Model building and refinement for cryo-EM structures**. The initial model of M4R was derived from the crystal structure of antagonist-bound M4R structure (PDB code, 5DSG)[29] and the mu-opioid receptor coordinate (PDB code, 6DDE)[36] was used as initial models for the $G_i$ and scFv16. UCSF Chimera[40] was used to

dock all the models into the EM density maps, and followed by iterative manual adjustment in Coot[41]. Agonist and PAM coordinates and geometry restrains were generated using Phenix.elbow[42]. The models were further refined in PHENIX[43]. Rosetta was used to determine if more optimal models existed. The geometries of models were validated using MolProbity[44]. Model overfitting was evaluated by refinement against one cryo-EM half map. Fourier shell correlations (FSCs) curves[45] were calculated on the basis of the final M4R–ligand–G$_i$–scFv16 model and the half map that was used for refinement, as well as the other half map that was used for cross-validation. Additionally, local resolutions were estimated using Blocres[39]. Structure figures were generated using PyMOL (http://www.pymol.org) and UCSF Chimera[40].

**Cyclic AMP accumulation assay**. To validate the effects of the mutations on the M4 G$_i$ protein signaling pathway, the split luciferase-based GloSensor cAMP biosensor technology (Promega) was used in this study. One day prior to assay, 1 μg M4 DNA and 1 μg pGloSensor™−22F cAMP Plasmid DNA (Promega) were co-transfected into HEK293T cells (ATCC CRL-11268; mycoplasma free) using Lipofectamine 2000 (Life Technologies). The cells grew for 4–5 h and were digested with trypsin. The cells were then added into 384-well white poly D-lysine-coated plates (Greiner) with Dulbecco's modified Eagle medium (DMEM; Life Technologies) supplemented with 1% dialyzed fetal bovine serum (dFBS) at a density of 10,000–15,000 cells in 40 μl medium per well and incubated overnight (20–24 h) at 37 °C in 5% carbon dioxide. The following day, the culture medium was removed from the cell plates. The wells were loaded with 20 μl 2 mg/ml D-luciferin sodium salt prepared in Hanks' balanced salt solution (HBSS) pH 7.4 and incubated for 1 h at 37 °C. All of the following steps were carried out at room temperature. To measure the activity of ACh, iperoxo or compound-110's activity at M4, 10 μl 4×ACh or iperoxo or compound-110 solution was added with a final concentration ranging from 1 nM to 3 μM and reacted for 15 min. The plates for the agonist assay were diluted by adding 10 μl isoproterenol (Sigma) at a final concentration of 200 nM, paused for 15 min and followed by measuring the luminescence using an Envision plate reader (Perkin Elmer). Data were analyzed using GraphPad Prism 8.1.

**Calcium mobilization assays**. To assess the potential agonists effects of compound-110 in G$_q$ signal pathway in muscarinic receptors, calcium mobilization assays with FLIPR$^{TETRA}$ were performed in HEK 293T cells (ATCC CRL-11268; mycoplasma free) for transient transfection of plasmids of wild type M1R, M3R, and M5R. Cells were cultured into 10-cm dishes and incubated overnight. Three dishes cells were transfected with 16 μl Lipofectamine 2000 (Life technologies) together with 4 μg wild type M1R, M3R, or M5R plasmid in 1 ml opti-MEM, separately. The plates were cultured for 6 h, cells were plated in the poly-L-lysine (PLL) coated 384-well plates at a density of 10–15,000 cells per well in DMEM containing 1% dialyzed FBS and incubated overnight.

The second day, medium was removed and cells were incubated with 20 μl/well Fluo-4 Direct dye, prepared in the assay buffer (1× HBSS, 2.5 mM probenecid, and 20 mM HEPES, pH 7.4) for 1 h at 37 °C, and followed 10 min incubation at room temperature in the dark. Cells were placed in a FLIPR$^{TETRA}$ fluorescence imaging plate reader (Molecular Devices) as the baseline before addition of 10 μl of 3× drug solutions, acetylcholine and compound-110, reconstituted at assay buffer and aliquoted into 384-well plates, separately. Fluorescence in each well was normalized to the average of the first 10 reads (one measurement read per second) (i.e., baseline fluorescence). Then, the maximum-fold increase, which occurred within 2 min after the agonist addition, over baseline fluorescence elicited was determined. Data were analyzed by GraphPad Prism 8.1.

**Bioluminescence resonance energy transfer assay**. The chimeric constructs (for example, GFP2-β-arrestin, Gα-RLuc8, and Gγ-GFP2 constructs) were designed and assays were performed as previously described[46]. Briefly, HEK 293T cells (ATCC CRL-11268; mycoplasma free) were plated either in 6-cm dishes at a density of 1–2 million cells per well, or 10-cm dishes at 3–4 million cells per dish. 12–14 h later, cells were transfected at the confluency of 60–80%, using a 1:1:1:1 DNA ratio of receptor: Gα-RLuc8: Gβ: Gγ-GFP2 (1 μg per construct for 6-cm dishes, 3 μg per construct for 10-cm dishes), or using a 1:1:10 DNA ratio of receptor-V2tail-RLuc8: GRK2: GFP2-β-arrestin (total DNA mass 4 μg for 6-cm dishes, 12 μg for 10-cm dishes). TransIT® 2020 (Mirus Biosciences) was used to complex the DNA at a ratio of 3 μl Transit per μg DNA, OptiMEM (Gibco) at a concentration of 10 ng DNA per μl OptiMEM. The next day, cells were reseeded in white opaque bottom 96-well assay plates (Beyotime) at a density of 30,000–50,000 cells per well. One day after plating in 96-well assay plates, the growth medium was carefully decanted and replaced immediately with 40 μl drug buffer (1×Hank's balanced salt solution (HBSS) + 20 mM HEPES, pH 7.4) containing 7.5 μM coelenterazine 400a (Nano-light Technologies). After a 2 min equilibration period, cells were treated with 20 μl compounds prepared in drug buffer at serial concentration gradient for an additional 5 min. Plates were then read in an LB940 Mithras plate reader (Berthold Technologies) with 395 nm (RLuc8-coelenterazine 400a) and 510 nm (GFP2) emission filters, at integration times of 1 s per well. BRET2 ratios were calculated as the ratio of the GFP2 emission to RLuc8 emission. Normalization was done using

the reference ligand (iperoxo) as a divisor, which means BRET2 ratio at [iperoxo]$_0$ was designated as 0%, BRET2 ratio at [iperoxo]$_{max}$ as 100%.

**Tango arrestin recruitment assay**. The receptor Tango constructs, which contain the TEV cleavage site and the tetracycline transactivator (tTA) fused to the C terminus of the receptor, were designed and assays were performed as previously described[47]. HTLA cells expressing TEV fused-β-Arrestin2 (provided by Richard Axel lab) and a tetracycline transactivator-driven luciferase were grown in HTLA media (10% FBS DMEM containing 5 mg/ml Puromycin and 100 mg/ml Hygromycin B). The day before transfection, HTLA cells were split to reach ~70% confluency the next day. The receptor Tango constructs were transfection at a ratio of 4 μl PEI per μg DNA (total DNA mass 3 μg for 6-cm dish). The next day, media and transfection reagents were removed, cells were washed with PBS, detached using trypsin, centrifuged, and resuspended in DMEM supplemented with 2% dialyzed FBS. Transfected cells were then plated onto poly-L-lysine-coated 384-well white clear bottom cell culture plates at a density of 10,000 cells/well in a total of 30 μl. The cells were incubated for at least 6 h before receiving drug stimulation to allow for recovery and adherence to the plate. Drug solutions were prepared in drug buffer (1×HBSS, 20 mM HEPES, 0.1% BSA, 0.01% ascorbic acid, pH 7.4) at 3× and added to cells (10 μl per well) for 16–20 h incubation. Drug solutions used for the Tango assay were exactly the same as used for the BRET2 assay, which was conducted in parallel to the Tango assay. After 16–20 h overnight incubation, media and drug solutions were removed from plates and 20 μl per well of BrightGlo reagent (purchased from Promega, after 1:20 dilution) was added per well. The plate was incubated for 20 min at room temperature in the dark before being counted using Envision or Ensight$^{TM}$ luminescence counter (PerkinElmer).

**Molecular docking**. The Schrodinger Suite 2019-4 (Schrödinger) was used to predict ligand binding poses for M4R. The structures of M4R were processed by using 'Protein Preparation Wizard'[48]. Ligands were processed by using 'LigPrep' (Schrödinger). 'Glide SP' was used for docking[49–51].

**Molecular dynamics simulation of M4R with iperoxo, PAM-LY2119620, and compound-110**. The M4R structures were isolated from their cryo-EM complex structures and Prime (Schrödinger) was used to add hydrogens, missing side chains, and cap the termini of the receptor. The residues D78$^{2.50}$ and D129$^{3.49}$ in M4R were manually protonated to simulate the protonation upon GPCR activation[52]. Then the processed M4R structure in complex with agonist (iperoxo/compound-110) and PAM LY2119620 was embedded in a bilayer composed of 148 1-palmitoyl-2-oleoyl-sn-glycero-3-phosphocholine (POPC) lipids using the CHARMM-GUI Membrane Builder. The orientation of M4R in the membrane is referenced to the M2R structure (PDB code: 6OIK) in the Orientations of Proteins in Membranes (OPM) database[53]. Each receptor-agonist-membrane system was solvated in a periodic 0.15 M NaCl TIP3P water box with a minimum water height of 20.0 Å on top and bottom of the system.

All simulations were performed on a GPU cluster using the CUDA version of PMEMD (Particle Mesh Ewald Molecular Dynamics) in Amber18 (AMBER 2018, University of California, San Francisco). The protein was modeled with the ff14SB protein force field[54], ligands with the GAFF2 force field[55] and lipids with the AMBER Lipid17 force field. The constructed system was firstly energy minimized for 10,000 steps, of which the first 5000 steps were performed using the steepest descent method and the remaining 5000 steps used the conjugate gradient method. Then the simulation system was heated from 0 to 100 K using Langevin dynamics with a constant box volume. Restraints were applied to protein, ligands, and lipids with a constant force of 10 kcal/mol/Å$^2$. Subsequently, the temperature was increased to 310 K, where the periodic box was coupled accordingly using anisotropic Berendsen control in order to maintain the pressure at around 1 atm. These restraints were then removed from lipids and the system was equilibrated for another 10 ns at the constant pressure and temperature ensemble (NPT). Further equilibration was then carried out at 310 K with harmonic restraints applied to the protein starting at 5 kcal/mol/Å$^2$ and reduced in a stepwise fashion every 2 ns for 10 ns, followed by 0.1 kcal/mol/Å$^2$ restraints for 20 ns for a total of 30 ns of equilibration. Then 1 μs production simulations with no restraints were performed at 310 K and 1 bar in the NPT ensemble for each system, and five independent runs (totally 5 μs/system) were performed. The particle mesh Ewald (PME)[56] method was used to treat all electrostatic interactions beyond a cutoff of 9 Å. The SHAKE algorithm[57] was used for recording the length of bonds involving hydrogen during the simulation with an integration time step of 2 fs.

Snapshots from each trajectory were saved every 40 ps during the production runs and these trajectories were used for analysis.

**Pharmacokinetics of compound-110**. Studies were conducted by Suzhou Kangrun Pharmaceutical Testing Service, Inc. (Suzhou, China). Male C57/BL6J mice (age 6–8 weeks, ~25 g body weight) were purchased from JOINN Laboratories, Inc. (Suzhou). For PK brain-penetration studies, compound-110 was dissolved in 5% DMSO + 95% saline, and administered at doses of 5 mg/kg (i.p.) with six animals in each group. Blood samples (0.1 ml) were collected from mouse orbit at 0.5 and 2 h, then centrifuged at 5000 rpm at 4 °C for 10 min to collect plasma samples. Brain tissues were collected at 0.5 and 2 h, washed with saline, weighed, and

homogenized in 50% cold methanol (brain weight(g)/50% methanol (ml) = 1/3) to obtain drug solutions. All samples were stored at −80 °C before analysis. Compound concentrations in the samples were determined using liquid chromatography–mass spectrometry (LC–MS/MS). All animal experiments were performed in accordance with the protocol (SIBCB-S375-1912-027) approved by the Animal Care and Use Committee of the Center for Excellence in Molecular Cell Science, Chinese Academy of Sciences.

**Locomotor activity**. Locomotor activity was assessed under standardized environmental conditions in 20 × 20-cm Plexiglas chambers. Mice movement was measured by an automated video tracking system (Etho Vision XT software; Noldus Information Technology). Male C57/BL6J mice (9–10 weeks old) were injected (i.p.) with vehicle (0.9% saline) or compound-110 (0.015625, 0.0625, 0.25, or 1 mg/kg) and placed into the open field. After 30 min, the NMDA receptor antagonist MK-801 (0.2 mg/kg), a psychotomimetic was administrated to induce hyperlocomotion and mice were immediately placed back to the open field. The activity was monitored throughout this entire period. The horizontal movement was measured as the total distance traveled in centimeters. The means ± SEMs of the locomotor responses were analyzed using Graphpad Prism 8.1. To estimate the half-maximal effective dose ($ED_{50}$), dose responses of total locomotor activity during the 45-min period after MK-801 administration were plotted and best-fit decay curves were determined using a nonlinear regression 'one-phase-exponential-decay' equation.

**Catalepsy**. Catalepsy was measured at 30 and 60 min after the administration of each test compound. Eight male C57/BL6J mice (9–10 weeks old) were used in each group. Measurements were performed three times at each observation time point by an observer blinded to the treatment. Mice were forced to hang by forepaw on the upper edge of a glass rod (diameter: 1 cm). A catalepsy response was recorded as the animals remained in the unnatural vertical position for 30 s or longer.

**Reporting summary**. Further information on research design is available in the Nature Research Reporting Summary linked to this article.

## Data availability

Data supporting the findings of this manuscript are available from the corresponding authors upon reasonable request. Source data are provided with this paper. The coordinates for M4R-c110-Gi-scFv16, M4R-ip-Gi-scFv16, and M4R-ip-LY-Gi-scFv16 have been deposited in the Protein Data Bank with the accession codes: 7V6A, 7V69, and 7V68. The EM maps for M4R-c110-Gi-scFv16, M4R-ip-Gi-scFv16, and M4R-ip-LY-Gi-scFv16 have been deposited in EMDB with the codes: EMD-31740, EMD-31739 and EMD-31738, respectively.

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

## Acknowledgements

This work was supported by the National Science Fund for Distinguished Young Scholars 32022038 (T.H.) and the National Natural Science Foundation of China (NSFC) 32071197 (S.W.), Thousand Talents Plan-Youth (S.W.) Shanghai Science and Technology Committee 19ZR1466200 (S.W.) and Shanghai Rising-Star Program 20QA1410600 (S.W.). We thank the Shanghai Municipal Government and ShanghaiTech University for financial support. The cryo-EM data were collected at the Bio-Electron Microscopy Facility, ShanghaiTech University, with the assistance of Q.-Q. Sun, D.D. Liu, and Y.H. Liu. We thank the Cloning, Cell Expression, Protein Purification and Assay Core Facilities of iHuman Institute for their support. Compound-110 was kindly provided by Merck.

## Author contributions

J.-J.W. designed the expression constructs, purified the M4R–Gi complexes, prepared the final samples for cryo-EM sample and cryo-EM data collection, and participated in the figure and manuscript preparation. M.W. performed the MD simulation and participated in the figure and manuscript preparation. Z.-C.C. performed the BRET and Tango assays, and participated in the figure and manuscript preparation. L.-J.W. EM data process and structure determination. T.W. assisted the cryo-EM sample preparation. D.-M.C. performed in vivo animal test. H.W. synthesized the compound-110 for animal studies. Y.-M.X. and F.L. performed the cAMP functional assay. S.-H.L. assisted with RMSF values calculation. J.-L.L. and N.C. assisted with the insect cell expression. S.-W.Z. supervised the MD simulation. J.-J.C. synthesized the compound-110 and manuscript editing. S.W. supervised the BRET and Tango assays, in vivo animal test and manuscript editing. T.H. conceived of the project and supervised the overall studies, analyzed the structures and wrote the manuscript.

## Competing interests

The authors declare no competing interests.
