## [Peer Review File · Nature Communications]

REVIEWER COMMENTS

Reviewer #1 (Remarks to the Author):

J. Wang et al. report the active structures of muscarinic receptor 4 (M4R) in complex with only iperoxo, iperoxo and the positive allosteric modulator LY2119620, and with the allosteric agonist compound110 by cryo-electron microscopy. From these models, they found different interaction modes and activation mechanisms. They also conducted molecular dynamics (MD) simulations to analyze the structures. Finally, they studied the effect of compound110 on schizophrenia using a schizophrenia-mimic mouse model, and showed that compound110 exhibits antipsychotic activities with low extrapyramidal side effects. Since the active structure of M4R, which is an important drug target for schizophrenia, has not been reported before, their work would be worthwhile for publication to Nature Communication. However, there are some major issues that should be revised in their work before publication.

Major issues:

- (i) Their included validation report from the wwPDB (attached pdf files) are only the pre-submission preliminary versions that are not meant peer review. The final version of the reports are required to adequately validate their EM structures.
- (ii) They executed MD simulations for three systems, M4R-iperoxo, M4R-iperoxo-LY2119620, and M4R-Compound110. Although their simulation length of 1 μ s during the production runs and the number of parallel runs per system (five independent runs, totaling 5 μ s/system) is enough, their data analyses are insufficient. E.g., on page 6 they write: we performed MD analysis of M4R with different ligands. The results show that iperoxo has lower flexibility in the M4R-ip-LY structure compared with that of M4R-ip structure (Extended Data Fig. 8a, b).
The RMSD values shown in the figure (8a/b) only shows the differences with respect to the corresponding experimental structure (also, these are averaged over all parallel trajectories, while it would be better to show each independent trajectory separately for such a graph). Instead of the RMSD, they should calculate the RMSF (root-mean-square fluctuation) to show the fluctuation of each ligand. In addition, the vertical axis titles in the extended data Fig. 8 are difficult to understand, so they should explain the meaning of 'Orth-iperox' and 'allo-LY' in the caption.
- (iii) The Discussion section is nothing more than a brief summary of their manuscript. At the very least, they should discuss what the 'unconventional activation' is, as they mention in the title of their manuscript, as well as the impact of their work on the greater scientific community.

Minor issue:

There are some typos such as 'otthosteric' and English grammar issues, so they should carefully revise their manuscript.

Reviewer #2 (Remarks to the Author):

Report on Wang et al, Nat Comms Muscarinic M4 structures

This is a nice paper reporting several structures of an important pharmaceutical drug target, in complexes with three different ligands, including an agonist, a positive allosteric modulator, and what the authors call an "allosteric agonist". This term "allosteric agonist" sounds very reasonable to me, but I am not trained in pharmacological terminology and definitions, so this terminology should be checked with a trained pharmacologist. The authors present an intelligent investigation of the interactions between the three ligands and the receptor, together with extensive discussions of how they influence the agonist-

induced structural rearrangements in the trans-membrane helical domain that couple to G protein activation. Overall, I think this paper would add significantly to the body of data on muscarinic GPCR structure and mechanism.

The clinical data shown in Fig. 5, as well as the paragraph entitled "investigation of antipsychotic potency of compound-110 in schizophrenia-mimic mouse model" are not so convincing, so I am neutral on whether these might be better to be deleted, leaving the manuscript to focus purely on the structures, structural biology and pharmacology. Similarly, the MD simulations shown in Extended Data Fig. 8 are not really all that convincing, nor are they necessary for the main conclusions of the paper.

Minor comments that require changes

I could not see the iperoxo compound in the left-hand structure in Fig. 1b, whereas I think I can see it in the right-hand structure that includes the additional ligand LY2119620. A molecular ball-and-stick model of iperoxo should be added to the left-hand depiction in Fig. 1b.

Reviewer #3 (Remarks to the Author):

The presented work "The Unconventional Activation of the Muscarinic Acetylcholine Receptor M4R by diverse ligands" presents agonist-specific conformations of M4 muscarinic acetylcholine receptor for agonist iperoxo, iperoxo and PAM LY2119620 and allegedly M4 selective agonist compound-110. In addition, the compound-110 exerts anti-psychotic effects without inducing catalepsy in the chemical model of schizophrenia.

Major points

It is known for a long time that structurally diverse agonists may stabilize agonist-specific conformations resulting in functional selectivity. This work is an opportunity to show molecular details of agonist-specific conformations of the M₄ receptor. However, I am afraid the authors missed their chance in this version of the manuscript. Conclusions on the crucial difference of the compound-110 action at the M₄ versus M₂ receptor are vague. The same applies to differences between iperoxo and the compound-110 at the M₄ receptor. The authors should be specific in their conclusions. The fundamental information is buried in the Results and not well supported by the data. Although the authors admit that the EM density does not allow for an unambiguous placement of iperoxo in the binding pocket, they report residues interacting with ligands and even the orientation of the side-chains. Although the resolution of reported structures is about 3.4 Å authors report differences among them as small as 1 Å. The authors should explain how they get such precise details of the structures.

The second critical issue of the work is the statement that compound-110 is selective for M₄ over M₂ and M₁ having potency about 20-fold higher for M₄ than for M₂ or M₁. First of all, the potency of an agonist is given by its affinity to the receptor (characteristic of ligand and receptor) and its operational efficacy (characteristic of receptor and system). The operational efficacy is affected by the system (subtype difference in the receptor coupling, expression level of individual constituents of the signalling pathway, etc.). Thus, the agonist potency is affected by the system in which it is measured. So, to compare potencies of the tested agonist at various receptor subtypes the potency of the tested agonist relative

to the potency of the reference agonist (e.g., acetylcholine) has to be taken (for details see for example Randakova and Jakubik 2021 Pharmacol Res 169:105641). Moreover, the compound-110 does not seem to possess potency in inhibiting cAMP formation or arrestin recruitment 20-fold higher at the M₄ receptor than at the M₂ receptor (Extended Data Fig. 1). The authors should compare potencies accordingly and re-evaluate their statements on compound-110 selectivity.

Specific minor comments

The authors state: *"The overall structures of the three complexes are comparable, with the root mean square deviations (RMSDs) of the Ca atoms of the receptors are around 0.9 Å." At 3.4 Å resolution, RMSD of 0.9 Å means that the structures are the same.*

The authors write: *"The diverse scales of TM6's outward movement may lead to the orientation differences of Gai subunits in those complex structures (Extended Data Fig. 5a)."* Could the authors be more specific about how big is the "difference"? Which agonist results in the bigger outward movement of the TM6?

The authors write: *"Generally, the binding interfaces of the three M4R and Gi complexes are more similar to that of M2R-GoA complex (Extended Data Fig. 5b, d)"* It is unclear which complexes are less similar than M2-GoA.

The authors write: *"Interestingly, the corresponding residue Pro181 in ECL2 of M4R does not form interactions with LY2119620 (Fig. 2c)."* What forms of interaction are theoretically possible for proline residues?

The authors write: *"The results show that iperoxo has lower flexibility in the M4R-ip-LY structure compared with that of M4R-ip structure (Extended Data Fig. 8a, b)."* It would be of interest to compare the data with iperoxo flexibility in 4MQS and 4MQT structures.

Language and typos

Subject and predicate mismatch occur several times in the manuscript.

It is somehow uncommon to describe the results in the present tense.

Saxon genitive is a possessive adjective and thus should be used with persons only.

Reference in "Compared to the inactive M4R crystal structure (ref?)," is missing.

Reference 20 and 52 are malformed.

REVIEWER COMMENTS

Reviewer #1 (Remarks to the Author):

J. Wang et al. report the active structures of muscarinic receptor 4 (M4R) in complex with only iperexo, iperexo and the positive allosteric modulator LY2119620, and with the allosteric agonist compound110 by cryo-electron microscopy. From these models, they found different interaction modes and activation mechanisms. They also conducted molecular dynamics (MD) simulations to analyze the structures. Finally, they studied the effect of compound110 on schizophrenia using a schizophrenia-mimic mouse model, and showed that compound110 exhibits antipsychotic activities with low extrapyramidal side effects. Since the active structure of M4R, which is an important drug target for schizophrenia, has not been reported before, their work would be worthwhile for publication to Nature Communication.

However, there are some major issues that should be revised in their work before publication.

Major issues:

(i) Their included validation report from the wwPDB (attached pdf files) are only the pre-submission preliminary versions that are not meant peer review. The final version of the reports are required to adequately validate their EM structures.

Response: Thanks for pointing this out. The coordinate deposition process was initiated in Aug of 2021 and the draft validation report was generated on Aug.18, 2021. So far, the deposition process is approved by PDB staff and will be released upon manuscript publication. The coordinate files and EM maps are available for reviewers, and the final validation reports are provided.

(ii) They executed MD simulations for three systems, M4R-iperexo, M4R-iperexo- LY2119620, and M4R-Compound110. Although their simulation length of 1 μ s during the production runs and the number of parallel runs per system (five independent runs, totaling 5 μ s/system) is enough, their data analyses are insufficient. E.g., on page 6 they write: we performed MD analysis of M4R with different ligands. The results show that iperexo has lower flexibility in the M4R-ip-LY structure compared with that of M4R-ip structure (Extended Data Fig. 8a, b). The RMSD values shown in the figure (8a/b) only shows the differences with respect to the corresponding experimental structure (also, these are averaged over all parallel trajectories, while it would be better to show each independent trajectory separately for such a graph). Instead of the RMSD, they should calculate the RMSF (root-mean-square fluctuation) to show the fluctuation of each ligand. In addition, the vertical axis titles in the extended data Fig. 8 are difficult to understand, so they should explain the meaning of 'Orth-iperox' and 'allo-LY' in the caption.

Response: Thanks for the suggestions, the Reviewer 2 also thinks that the MD simulation is not much relevant to the main conclusion of this manuscript. Even our intention on using MD simulation is to confirm the cooperativity between iperexo and LY2119620 in M4R-iperexo- LY2119620 complex structure as reported previous study (Croy, C.H. et al, Mol Pharmacol 86, 106-115). At the same time, the agreement

between experimental and calculation data may further validate our structure determination results. However, we agree with both reviewers that the MD simulation may dilute the focus of audiences on the main theme of the manuscript. We thus have decided to remove Extended Data Fig.8. in the revised manuscript.

(iii) The Discussion section is nothing more than a brief summary of their manuscript. At the very least, they should discuss what the ‘unconventional activation’ is, as they mention in the title of their manuscript, as well as the impact of their work on the greater scientific community.

Response: Thanks for the suggestion. We have revised the “**Discussion**” in the revised manuscript.

Minor issue:

There are some typos such as ‘othosteric’ and English grammar issues, so they should carefully revise their manuscript.

Response: Thanks for pointing this out. We have revised the manuscript carefully and corrected the spelling mistakes.

Reviewer #2 (Remarks to the Author):

Report on Wang et al, Nat Comms Muscarinic M4 structures

This is a nice paper reporting several structures of an important pharmaceutical drug target, in complexes with three different ligands, including an agonist, a positive allosteric modulator, and what the authors call an “allosteric agonist”. This term “allosteric agonist” sounds very reasonable to me, but I am not trained in pharmacological terminology and definitions, so this terminology should be checked with a trained pharmacologist. The authors present an intelligent investigation of the interactions between the three ligands and the receptor, together with extensive discussions of how they influence the agonist-induced structural rearrangements in the trans-membrane helical domain that couple to G protein activation. Overall, I think this paper would add significantly to the body of data on muscarinic GPCR structure and mechanism.

We thank the reviewer’s positive comments on our study.

The clinical data shown in Fig. 5, as well as the paragraph entitled “investigation of antipsychotic potency of compound-110 in schizophrenia-mimic mouse model” are not so convincing, so I am neutral on whether these might be better to be deleted, leaving the manuscript to focus purely on the structures, structural biology and pharmacology.

Response: Thanks for the reviewer’s comment. Most anti-schizophrenia drugs targeting on M1R/M4R show dose-limiting side effects due to non-selective activation on other peripheral mAChR subtypes. The

binding position and activation mechanism of compound-110 in M4R are unique, its effect in the schizophrenia-mimic mouse model is an important indicator of its clinical development potentials. Fortunately, the results are promising. So, we'd like to keep this animal data, while also adding the limitation of the compound-110 from the animal experiments in the revised manuscript. At the same time, we rephrased the description in “**Abstract**” to make the manuscript more focused on structures.

Similarly, the MD simulations shown in Extended Data Fig. 8 are not really all that convincing, nor are they necessary for the main conclusions of the paper.

Response: Agreed with the reviewer's suggestion, the Extended Data Fig. 8 has been deleted in the revised manuscript.

Minor comments that require changes

I could not see the iperoxo compound in the left-hand structure in Fig. 1b, whereas I think I can see it in the right-hand structure that includes the additional ligand LY2119620. A molecular ball-and-stick model of iperoxo should be added to the left-hand depiction in Fig. 1b.

Response: As the electron density map of iperoxo is not clear enough in the M4R-iperoxo complex structure, we decided rather not model iperoxo in the structure with low confidence. This phenomenon is similar with that in M1-iperoxo-G11 structure (PDB code 6OIJ, *Science*. 2019; 364(6440): 552–557) where the density of iperoxo is poor. Fortunately, due to the cooperative influence of LY2119620, the improved density for iperoxo allowed us to put iperoxo into the M4R-iperoxo- LY2119620 complex structure unambiguously which is shown in the right-hand in Fig. 1b.

Reviewer #3 (Remarks to the Author):

The presented work “The Unconventional Activation of the Muscarinic Acetylcholine Receptor M4R by diverse ligands” presents agonist-specific conformations of M4 muscarinic acetylcholine receptor for agonist iperoxo, iperoxo and PAM LY2119620 and allegedly M4 selective agonist compound-110. In addition, the compound-110 exerts anti-psychotic effects without inducing catalepsy in the chemical model of schizophrenia.

Major points

It is known for a long time that structurally diverse agonists may stabilize agonist-specific conformations resulting in functional selectivity. This work is an opportunity to show molecular details of agonist-

specific conformations of the M₄ receptor. However, I am afraid the authors missed their chance in this version of the manuscript.

Response: Agree with the reviewer's point that this study is an excellent example to show how diverse ligands binding leads to different functional selectivity. However, the fact is not as simple as we thought because there is no obvious difference from the structure framework point of view when comparing those three structures, as stated in the manuscript. As a matter of fact, the real difference is mostly at the atomic level which lies in the ligand binding mode (position and orientation), ligand-residue sidechain interactions. All those comparative difference analyses have been illustrated in the sessions of **“Activation mechanism of M4R by iperoxo and LY2119620”**, **“The binding mode of compound-110 in M4R”** and **“The activation features of M4R by compound-110”**. Thus, if we have to make conclusions on this aspect, we would speculate the functional selectivity is in micro scale.

Conclusions on the crucial difference of the compound-110 action at the M₄ versus M₂ receptor are vague. The same applies to differences between iperoxo and the compound-110 at the M₄ receptor. The authors should be specific in their conclusions. The fundamental information is buried in the Results and not well supported by the data. Although the authors admit that the EM density does not allow for an unambiguous placement of iperoxo in the binding pocket, they report residues interacting with ligands and even the orientation of the side-chains. Although the resolution of reported structures is about 3.4 Å authors report differences among them as small as 1 Å. The authors should explain how they get such precise details of the structures.

Response: Thanks for the reviewer's comments. Structure comparison analysis has been described between M4 and M2 complex. However, we could not make comparative analysis between compound-110 activated M4 and M2 receptors due to the lack of M2-compound-110 complex structure.

As addressed in the previous question. All comparative difference analysis has been illustrated in the sessions of **“Activation mechanism of M4R by iperoxo and LY2119620”**, **“The binding mode of compound-110 in M4R”** and **“The activation features of M4R by compound-110”**.

Additionally, we have rephrased some conclusions according to the data and reviewer's suggestions.

Regarding to the concern of “3.4 Å resolution structure cannot tell as small as 1 Å differences”, well, here probably is a misunderstanding on the resolutions in crystal structure and cryo-EM structure. The resolution in crystal structure is diffraction resolution which has physical meaning of crystal diffracting X-rays. However, the resolution in cryo-EM structure is called nominal or global resolution. In cryo-EM structures, the resolution at different portions of structure is different. For example, a 3.4 Å nominal resolution may exhibit better than 3 Å resolution at the certain place or low than 3.4 Å resolution in the other place of the structure.

The second critical issue of the work is the statement that compound-110 is selective for M₄ over M₂ and M₁ having potency about 20-fold higher for M₄ than for M₂ or M₁. First of all, the potency of an agonist is given by its affinity to the receptor (characteristic of ligand and receptor) and its operational efficacy (characteristic of receptor and system). The operational efficacy is affected by the system (subtype difference in the receptor coupling, expression level of individual constituents of the signalling pathway, etc.). Thus, the agonist potency is affected by the system in which it is measured. So, to compare potencies of the tested agonist at various receptor subtypes the potency of the tested agonist relative to the potency of the reference agonist (e.g., acetylcholine) has to be taken (for details see for example Randakova and Jakubik 2021 Pharmacol Res 169:105641). Moreover, the compound-110 does not seem to possess potency in inhibiting cAMP formation or arrestin recruitment 20-fold higher at the M₄ receptor than at the M₂ receptor (Extended Data Fig. 1). The authors should compare potencies accordingly and re-evaluate their statements on compound-110 selectivity.

Response: Thanks for pointing this out, after careful inspection of our data and the literatures, we have rewritten this portion as shown at line 1-5 on page 4 of the revised manuscript. In addition, we have included the activation curves of the reference ligand (acetylcholine) in the revised Extended Data Fig. 1b.

Specific minor comments

The authors state: “The overall structures of the three complexes are comparable, with the root mean square deviations (RMSDs) of the Ca atoms of the receptors are around 0.9 Å.” At 3.4 Å resolution, RMSD of 0.9 Å means that the structures are **the same**.

Response: Agree with the reviewer, we have revised the statement to “The overall structures of the three complexes are similar, with the root mean square deviations (RMSDs) of the Ca atoms of the receptors are around 0.9 Å”.

The authors write: “*The diverse scales of TM6’s outward movement may lead to the orientation differences of Gai subunits in those complex structures (Extended Data Fig. 5a).*” Could the authors be more specific about how big is the "difference"? Which agonist results in the bigger outward movement of the TM6?

Response: We have revised it to “*The diverse scales of TM6’s outward movement may lead to the orientation differences of Gai subunits in those complex structures, where M4R-ip-Gi, M4R-ip-LY-Gi and M4R-c110-Gi structures show the outward movement of 10.5 Å, 11.4 Å and 12.3 Å, respectively, compared with the inactive state of M4R structure (Extended Data Fig. 5a)*”.

The authors write: “*Generally, the binding interfaces of the three M4R and Gi complexes are more similar to that of M2R-GoA complex (Extended Data Fig. 5b, d)*” It is unclear which complexes are less similar than M2-GoA.

Response: We have revised it to “*Generally, the binding interfaces of the three M4R and Gi complexes are similar to that of M2R-GoA complex (Extended Data Fig. 5b, d)*”.

The authors write: “*Interestingly, the corresponding residue Pro181 in ECL2 of M4R does not form interactions with LY2119620 (Fig. 2c).*” What forms of interaction are theoretically possible for proline residues?

Response: Thanks for pointing this out. We have rephrased it to “*Interestingly, the corresponding residue in ECL2 is Pro181 in M4R and it does not form interactions with LY2119620 (Fig. 2c).*”

The authors write: “*The results show that iperexo has lower flexibility in the M4R-ip-LY structure compared with that of M4R-ip structure (Extended Data Fig. 8a, b).*” It would be of interest to compare the data with iperexo flexibility in 4MQS and 4MQT structures.

Response: Since the MD simulation study has been removed and the flexibility discussion on iperexo is also deleted.

Language and typos

Subject and predicate mismatch occur several times in the manuscript.

Response: Corrected.

It is somehow uncommon to describe the results in the present tense.

Response: Corrected.

Saxon genitive is a possessive adjective and thus should be used with persons only.

Response: Corrected.

Reference in “Compared to the inactive M4R crystal structure (ref?),” is missing.

Response: Added.

Reference 20 and 52 are malformed.

Response: Corrected.

REVIEWER COMMENTS

Reviewer #1 (Remarks to the Author):

I have a major issue with the management of their molecular dynamics (MD) simulation data. The authors deleted the below sentence from their original manuscript without any obvious reason. For a better understanding of the allosteric modulation or cooperativity between iperoxo and LY2119620, we performed MD analysis of M4R with different ligands. The results show that iperoxo has lower flexibility in the M4R-ip-LY structure compared with that of M4R-ip structure (Extended Data Fig. 8a, b).

However, they kept the following sentence in the current version of their manuscript: In our MD simulation of M4R-c110 structure, Tyr4166.51 is observed to maintain a relatively stable distance with Trp4136.48, which facilitates a stable interaction (Extended Data Fig. 10).

At the very least, they should have replied why their MD data cannot explain the allosteric modulation or cooperativity between iperoxo and LY2119620. If there is something wrong with their MD simulations, they should delete everything related to their MD simulations or redo them properly. However, if something is wrong with their reported structures, this manuscript is worthless. E.g., a quick check of their structures (based on the validation reports), suggests a large number of clashes and some issues with the ligand geometry of 5XI, IXO and 2CU. Furthermore, comparing the overall quality of the structures with the recent EM structures by Draper-Joyce (10.1038/s41586-021-03897-2, e.g. PDB ID 7ld3), shows that the structures by Wang et al are clearly inferior. Furthermore, the still lacking discussion section, the abolishment of the MD data compounded by the poor response from the authors weakens the impact of their manuscript further, so that their manuscript is not suitable for Nature Communications.

Reviewer #3 (Remarks to the Author):

The Authors addressed all my concerns except to one point to which they responded: *Regarding to the concern of "3.4 Å resolution structure cannot tell as small as 1 Å differences", well, here probably is a misunderstanding on the resolutions in crystal structure and cryo-EM structure. The resolution in crystal structure is diffraction resolution which has physical meaning of crystal diffracting X-rays. However, the resolution in cryo-EM structure is called nominal or global resolution. In cryo-EM structures, the resolution at different portions of structure is different. For example, a 3.4 Å nominal resolution may exhibit better than 3 Å resolution at the certain place or low than 3.4 Å resolution in the other place of the structure.*

I am afraid the problem persists. Reported "resolution" in databases (e.g., RCSB) is global resolution both for cryo-EM and X-ray structures. Authors' "nominal resolution" is the most likely "coordinate error" (a precision estimate in three dimensions (x, y, z) of the coordinates for each atom in the model) that indeed varies among the atoms and through the structure. However, it hardly surpasses the double accuracy of global resolution. Thus, in this case, everything less than 1.5 Å could be deemed below coordinate error (the same) unless the calculation of the localized correlation coefficient between measured data and data predicted from the model proves otherwise.

The manuscript improved substantially, however, there is still room for improvement. Eg.:

Ln 52 - "M4R relative selective allosteric agonist" -> "M4R-preferring allosteric agonist"

Ln 71 - "an agonist of M4R with small selectivity on potency over M2R and M1R" -> "an agonist with slightly higher potency at M4R than at M2R and M1R"

The authors also introduced new typos with revision.

REVIEWER COMMENTS

Reviewer #1 (Remarks to the Author):

I have a major issue with the management of their molecular dynamics (MD) simulation data. The authors deleted the below sentence from their original manuscript without any obvious reason.

For a better understanding of the allosteric modulation or cooperativity between iperoxo and LY2119620, we performed MD analysis of M4R with different ligands. The results show that iperoxo has lower flexibility in the M4R-ip-LY structure compared with that of M4R-ip structure (Extended Data Fig. 8a, b). However, they kept the following sentence in the current version of their manuscript:

In our MD simulation of M4R-c110 structure, Tyr4166.51 is observed to maintain a relatively stable distance with Trp4136.48, which facilitates a stable interaction (Extended Data Fig. 10).

At the very least, they should have replied why their MD data cannot explain the allosteric modulation or cooperativity between iperoxo and LY2119620. If there is something wrong with their MD simulations, they should delete everything related to their MD simulations or redo them properly. However, if something is wrong with their reported structures, this manuscript is worthless. E.g., a quick check of their structures (based on the validation reports), suggests a large number of clashes and some issues with the ligand geometry of 5XI, IXO and 2CU. Furthermore, comparing the overall quality of the structures with the recent EM structures by Draper-Joyce (10.1038/s41586-021-03897-2, e.g. PDB ID 7ld3), shows that the structures by Wang et al are clearly inferior. Furthermore, the still lacking discussion section, the abolishment of the MD data compounded by the poor response from the authors weakens the impact of their manuscript further, so that their manuscript is not suitable for Nature Communications.

Response: We thank the reviewer for the detailed evaluation of our manuscript and we are also thankful for the critical questions and comments. We are happy to take the opportunity to improve the manuscript. However, there are some misunderstanding as well. Here are the answers:

Concerning the removal of MD data, as a matter of fact, we had already prepared the responses to Reviewer #1's questions concerning our MD work. Following his suggestion, we had revised the MD related figures (Extended Data Fig. 8a, b) by showing each independent MD trajectory separately, changed the vertical axis titles. In addition, we had added a new panel replacing the old Extended Data Fig. 8b to show the RMSF values of iperoxo molecule in those three MD runs, which was also suggested by Reviewer #1.

However, later on, we had this part of data (including Extended Data Fig. 8a, b) removed from the last submission because Reviewer #2 thinks the MD simulation is not much relevant to this study and suggested removal. Since Reviewer #1 wants to see our answers to the questions on MD, plus, we also think that the MD simulations validated the cooperativity between iperoxo and LY2119620 pretty well. Thus, we'd like

to keep the revised Extended Data Fig. 8a and the new Extended Data Fig. 8b on RMSF in this revised version.

BTW, the removal of the MD data was not because there is anything wrong with the MD. Just the opposite, our MD results agree with our experimental data very well, which is that there is cooperativity between iperoxo and LY2119620 in M4R-ip-LY-Gi structure. Even in our original Fig. 8a, the RMSD fluctuation of iperoxo in M4R-ip-Gi structure is obviously larger than that of iperoxo in M4R-ip-LY-Gi structure as shown below, indicating that LY2119620 helps to stabilize iperoxo's binding to the receptor. However, we agree with Reviewer #1's suggestion, showing the RMSD in three independent trajectories is more convincing, RMSF is also more applicable in this case.

Updated Extended Data Fig. 8

Regarding to Reviewer #1's comment: "However, if something is wrong with their reported structures, this manuscript is **worthless**. E.g., a quick check of their structures (based on the validation reports), suggests a large number of clashes and some issues with the ligand geometry of 5XI, IXO and 2CU. Furthermore, comparing the overall quality of the structures with the recent EM structures by Draper-Joyce (10.1038/s41586-021-03897-2, e.g. PDB ID 7ld3), shows that the structures by Wang et al are **clearly inferior**", we disagree with Reviewer #1's comments that our **manuscript is worthless** or our **structures are inferior**. First of all, our manuscript reports three structures of M4 receptor complexes with G protein and different ligands, including the new type of compound-110 with novel modulation mode. These structures have never been reported before. Furthermore, we uncovered the structural basis of compound-110's allosteric selectivity and agonist profile, which is novel comparing with previous reported mAChRs

structures. As for the structure quality, the nature of different receptors is different, the different modulation ligands may also result receptors into diverse conformational stability. It is unfortunate that the conformation of M4 receptor in our case of ligands modulation is more flexible, which resulted lower resolution structures. For example, our structures were determined at 3.4 Å and 3.6 Å resolution, separately, while the adenosine A1-Gi2 structure was solved at higher resolution of 3.2 Å. However, it is not fair to judge the value of a research work simply from the apparent parameters, such as resolution. We think the true value of a research work is from the novelty, if the experimental evidence can support the conclusions, and if the claim can be cross validated. **In our case, our data is sufficient to backing up our conclusions.** Never the less, we have further improved the structure quality and reduce the clashes in the new deposited coordinates. The overall new clash scores and the geometry of the ligands are significantly reduced as shown in the new PDB deposition validation reports.

Reviewer #3 (Remarks to the Author):

The Authors addressed all my concerns except to one point to which they responded:

Regarding to the concern of “3.4 Å resolution structure cannot tell as small as 1 Å differences”, well, here probably is a misunderstanding on the resolutions in crystal structure and cryo-EM structure. The resolution in crystal structure is diffraction resolution which has physical meaning of crystal diffracting X-rays. However, the resolution in cryo-EM structure is called nominal or global resolution. In cryo-EM structures, the resolution at different portions of structure is different. For example, a 3.4 Å nominal resolution may exhibit better than 3 Å resolution at the certain place or low than 3.4 Å resolution in the other place of the structure.

I am afraid the problem persists. Reported “resolution” in databases (e.g., RCSB) is global resolution both for cryo-EM and X-ray structures. Authors’ “nominal resolution” is the most likely “coordinate error” (a precision estimate in three dimensions (x, y, z) of the coordinates for each atom in the model) that indeed varies among the atoms and through the structure. However, it hardly surpasses the double accuracy of global resolution. Thus, in this case, everything less than 1.5 Å could be deemed below coordinate error (the same) unless the calculation of the localized correlation coefficient between measured data and data predicted from the model proves otherwise.

The manuscript improved substantially, however, there is still room for improvement. Eg.:

Ln 52 - “M4R relative selective allosteric agonist” -> “M4R-preferring allosteric agonist”

Ln 71 - “an agonist of M4R with small selectivity on potency over M2R and M1R” -> “an agonist with slightly higher potency at M4R than at M2R and M1R”

The authors also introduced new typos with revision.

Response: We thank Reviewer #3’s patience and careful inspection of the revised manuscript. We have corrected the typos and some other incorrect expressions pointed out by Reviewer #3. In addition, We are sorry for still not being able to explain the resolution issue clearly. We agree with Reviewer #3’s argument

on the limitations from resolutions of cryo-EM maps. What we are trying to express is that, in the case of cryo-EM structure determination endeavor, there are more factors affecting the model quality than just the global resolution. For example, the stereo chemistry knowledge, such as bond length, bonds angle and dihedral angle, etc. may improve the structure quality. Anyhow, we don't intend to complicate the discussion any further and there is not serious disagreement between us and Reviewer #3. We really appreciate Reviewer #3's patience and devotion to this review. In addition, we have further improved the structure quality and reduce the clashes in the new deposited coordinates. The overall new clash scores are significantly reduced as shown in the new PDB deposition reports.

REVIEWERS' COMMENTS

Reviewer #1 (Remarks to the Author):

This manuscript is acceptable with a minor revision. Their Abstract section should be revised by adding the PAM's cooperativity as they mentioned in the first paragraph of the Discussion section.

REVIEWER COMMENTS

Reviewer #1 (Remarks to the Author):

This manuscript is acceptable with a minor revision. Their Abstract section should be revised by adding the PAM's cooperativity as they mentioned in the first paragraph of the Discussion section.

Response: We thank the reviewer for the detailed evaluation of our manuscript, we have added "*and the M4R-ip-LY-G_i structure validates the cooperativity between iperexo and LY2119620 on M4R.*" in "**Abstract**" in the revised manuscript.